# Maximum knee extension velocity without external load is a stronger determinant of gait function than quadriceps strength in the early postoperative period following total knee arthroplasty

Akira Iwata[1]*, Yuki Sano[2], Hideyuki Wanaka[2], Shingo Kobayashi[3], Kensuke Okamoto[3], Jun Yamahara[4], Masaki Inaba[4], Yuya Konishi[5], Junji Inoue[6], Atsuki Kanayama[6], Saki Yamamoto[1], Hiroshi Iwata[7]

1 Graduate School of Rehabilitation Science, Osaka Metropolitan University, Habkino, Osaka, Japan, 2 Department of Rehabilitation, Osaka General Medical Center, Osaka, Osaka, Japan, 3 Department of Rehabilitation, Osaka Rosai Hospital, Sakai, Osaka, Japan, 4 Department of Rehabilitation, Osaka Minami Medical Center, Kawachinagano, Osaka, Japan, 5 Department of Rehabilitation, Saiseikai Suita Hospital, Suita, Osaka, Japan, 6 Department of Physical Therapy, Faculty of Comprehensive Rehabilitation, Osaka Prefecture University, Habkino, Osaka, Japan, 7 Department of Cardiovascular Medicine, Juntendo University Graduate School of Medicine, Bunkyo, Tokyo, Japan

* iwata@omu.ac.jp

## Abstract

### Objective

Quadriceps weakness is considered the primary determinant of gait function after total knee arthroplasty (TKA). However, many patients have shown a gap in improvement trends between gait function and quadriceps strength in clinical situations. Factors other than quadriceps strength in the recovery of gait function after TKA may be essential factors. Because muscle power is a more influential determinant of gait function than muscle strength, the maximum knee extension velocity without external load may be a critical parameter of gait function in patients with TKA. This study aimed to identify the importance of knee extension velocity in determining the gait function early after TKA by comparing the quadriceps strength.

### Methods

This prospective observational study was conducted in four acute care hospitals. Patients scheduled for unilateral TKA were recruited (n = 186; age, 75.9 ± 6.6 years; 43 males and 143 females). Knee extension velocity was defined as the angular velocity of knee extension without external load as quickly as possible in a seated position. Bilateral knee function (knee extension velocity and quadriceps strength), lateral knee function (pain and range of motion), and gait function (gait speed and Timed Up and Go test (TUG)) were evaluated before and at 2 and 3 weeks after TKA.

**Data Availability Statement:** All relevant data are within the manuscript and its Supporting Information files.

**Funding:** This work was supported by Japan Society for the Promotion of Science KAKENHI Grant Numbers 20K11162. The funders had no role in study design, data collection, and analysis, decision to publish or preparation of the manuscript.

**Competing interests:** The authors have declared that no competing interests exist.

## Results

Both bilateral knee extension velocities and bilateral quadriceps strengths were significantly correlated with gait function. The knee extension velocity on the operation side was the strongest predictor of gait function at all time points in multiple regression analysis.

## Conclusion

These findings identified knee extension velocity on the operation side to be a more influential determinant of gait function than impairments in quadriceps strength. Therefore, training that focuses on knee extension velocity may be recommended as part of the rehabilitation program in the early postoperative period following TKA.

## Trial registration

UMIN Clinical Trials Registry (UMIN-CTR) UMIN000020036.

## Introduction

Knee osteoarthritis (OA) is one of the most common musculoskeletal disorders and is a leading cause of disability in older people [1]. Total knee arthroplasty (TKA) is the most common surgical intervention in individuals with end-stage knee OA [2]. TKA provides significant benefits concerning disease-specific and generic health-related quality of life, leading to patient satisfaction [3]. However, it is not uncommon that the decline in gait function continues after TKA surgery. Previous studies have demonstrated that it takes a few months for gait function to recover to the preoperative level [4, 5], and improvement to the same level as healthy older people is difficult even years after TKA [6, 7].

Quadriceps strength has been shown to affect the biomechanics (kinematics and kinetics) of the knee joint during gait in patients with TKA [8, 9] and has been significantly correlated with gait function [10–13]. Furthermore, several intervention studies have revealed that the rehabilitation programs focused on quadriceps strength effectively improve gait function in patients who undergo TKA [14–17]. As these studies have shown, quadriceps weakness has been considered a significant contributor to gait function decline after TKA.

However, some studies reported a large gap between the reduction rate in gait function and the decrease rate in quadriceps strength following TKA [4, 10, 18]. For example, in a previous study we reported that the decline in strength (44%) was far removed from the reduction in gait speed (8%) at three weeks after the surgery [18]. Based on these facts, the improvement in gait function after TKA may be affected by knee function other than quadriceps strength.

Muscle power is more of an essential determinant of gait function than muscle strength in community-dwelling older adults [19, 20]. Similarly, in patients who underwent TKA, gait speed was more closely associated with leg press power than with quadriceps strength [12]. Muscle power is a product of muscle strength and movement velocity; therefore, the movement velocity could also be considered a significant determinant of the gait function. Movement velocity is the speed at which the upper [21] /lower limbs [22–24] or trunk [25, 26] are moved as quickly as possible under various loading conditions. Previous studies have shown that the movement velocity of the lower limbs was an essential predictor of gait function in both community-dwelling [22] and institutionalized older adults [23]. Furthermore, the movement velocity was an increasingly important component of functional ability with aging and mobility limitation [27].

These facts suggest that the movement velocity of knee extension (knee extension velocity) is possibly a more critical parameter than quadriceps strength in determining the gait function in patients who undergo TKA. This study aimed to identify the importance of knee extension velocity in determining the gait function early after TKA by comparing the quadriceps strength of the study participants.

## Materials and methods

### Participants

This prospective observational study was conducted between June 2015 and January 2017 in four acute care hospitals. Patients scheduled to undergo unilateral TKA were recruited for the study. The inclusion criteria were as follows: (1) diagnosed knee OA, (2) primary unilateral TKA (3) age $\geq$ 60 years, (4) ability to walk for 10 m independently, with or without a cane, and (5) ability to understand and follow instructions.

This study was approved by the appropriate ethics committees of Osaka Prefecture University (approval No. 2015–105), and written informed consent was obtained from all participants. This study was registered in the University Hospital Medical Information Network Clinical Trials Registry (UMIN-CTR; UMIN000020036).

### Measurements

We collected demographic information, including age, sex, height, weight, and body mass index (BMI) for each participant. Clinical data were prospectively collected preoperatively and at 2 and 3 weeks postoperatively. We measured maximum gait speed and TUG to assess the gait function. TUG is a reliable and valid test to quantify functional mobility in older adults [28]. We measured the knee extension velocity, quadriceps strength, passive range of motion (ROM), and pain to evaluate knee function.

**Gait function.**   We measured maximum gait speed on an 8-m walkway in the rehabilitation room. The initial and final 1.5 m were excluded to allow for acceleration and deceleration. Patients were instructed to walk as fast as possible, and the time taken to walk the middle 5 m of the walkway was measured using a stopwatch. Gait speed was expressed in meters per second (m/s). A cane was provided for safety in a few patients at the discretion of the attending physical therapist. Gait speed was measured twice, and the higher value was used in the analyses.

TUG was performed as described by Podsiadlo and Richardson [28]. The patients were asked to stand up from a chair, walk 3 m at their usual pace, cross a line, turn around, walk back, and sit down. The time required to complete TUG was measured in seconds (sec) using a stopwatch. Each patient performed the test twice, and the faster time was used in the analyses.

**Knee function.**   Knee extension velocity was defined as the angular velocity of knee extension. Fig 1 shows the apparatus used to measure knee extension velocity. The subject was seated on an elevated bed with a wireless gyro-sensor (MVP-RF8-HC, W45×D45×H18.5 mm, 60 g, MicroStone, Nagano, Japan) on the distal lower limb. The sampling rate of the gyro-sensor was 200Hz and the measurement data were transferred to a personal computer. We made a target by wrapping a soft cushion on the frame, and it was set such that the center of the lower leg should hit it 160˚ of the knee joint angle. The patient was then instructed to extend the knee joint as quickly as possible from 90˚ of knee joint angle to hit the target without any external load. We measured the angular velocity of knee extension in degrees per second (˚/s). This test was performed five times after one practice trial, and the best result was used for subsequent analyses.

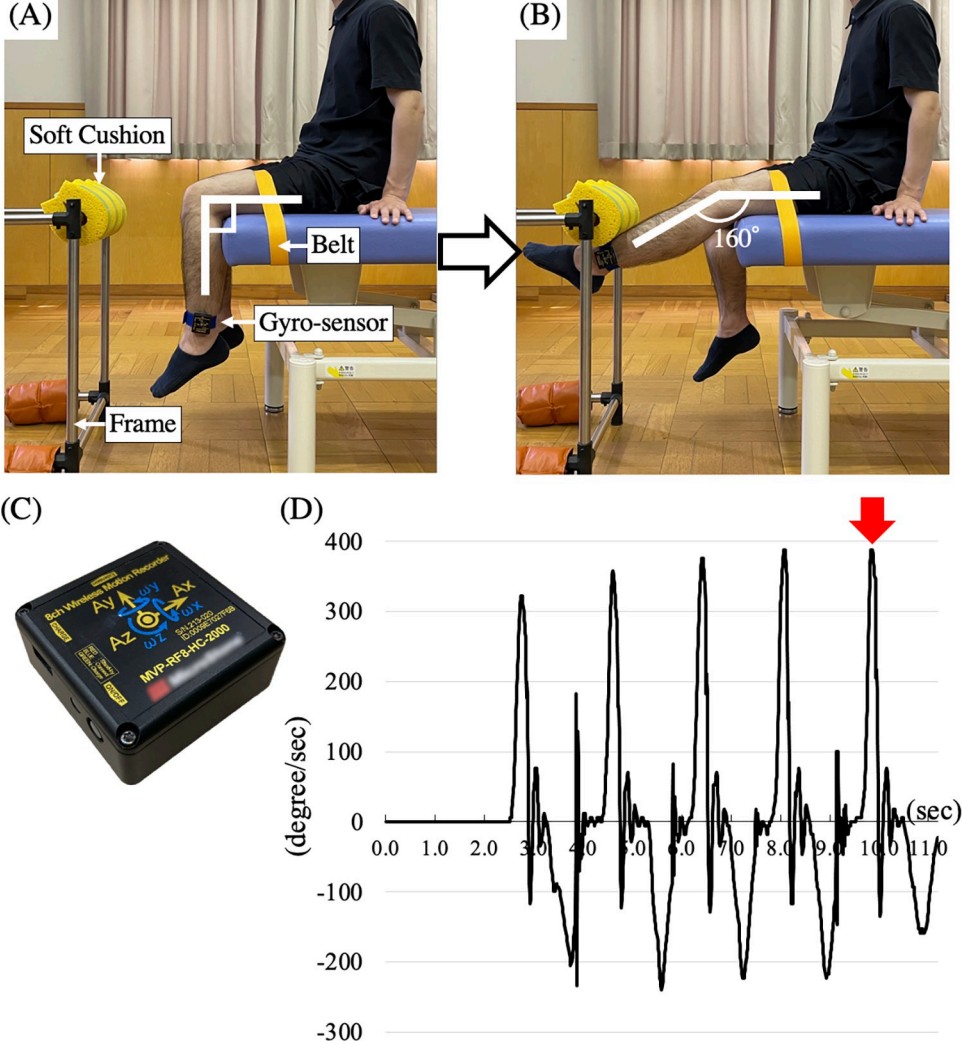

**Fig 1. Measurement of the maximum knee extension velocity.** (A, B) Experimental setup for measurement. The patient was seated on an elevated bed with a wireless gyro-sensor, where they extended their knee joint five times as quickly as possible from 90˚(A) to 160˚(B) to hit the soft cushion. (C) The gyro-sensor. (D) Representative data of knee extension velocity. The red arrow indicates the maximum knee extension velocity of five times.

Quadriceps muscle strength was defined as a voluntary isometric knee extension force. We measured the strength of both the operated and non-operated limbs. Fig 2 shows the setup for measuring the quadriceps strength. We used a hand-held dynamometer (HDD) with a fixing belt (μ-Tas F-1, Anima Corp., Tokyo, Japan). The patient was seated upright with their arms crossed. The sensor was set in the dynamometer in front of the distal leg with a belt. The patient was instructed to extend the knee with as much force as possible for 3 s, and the strength was measured in kilogram (kg). The knee joint angle was set at 110˚ during the measurement. To match the knee joint angle to 110˚, the measurer adjusted the length of the belt during a practice trial. This test was performed twice after one practice trial, and the larger value was used for the subsequent analyses.

Passive knee flexion and extension ROMs were measured using a goniometer in degrees (˚) with the patient in the supine position and recorded to the nearest 5˚ [29].

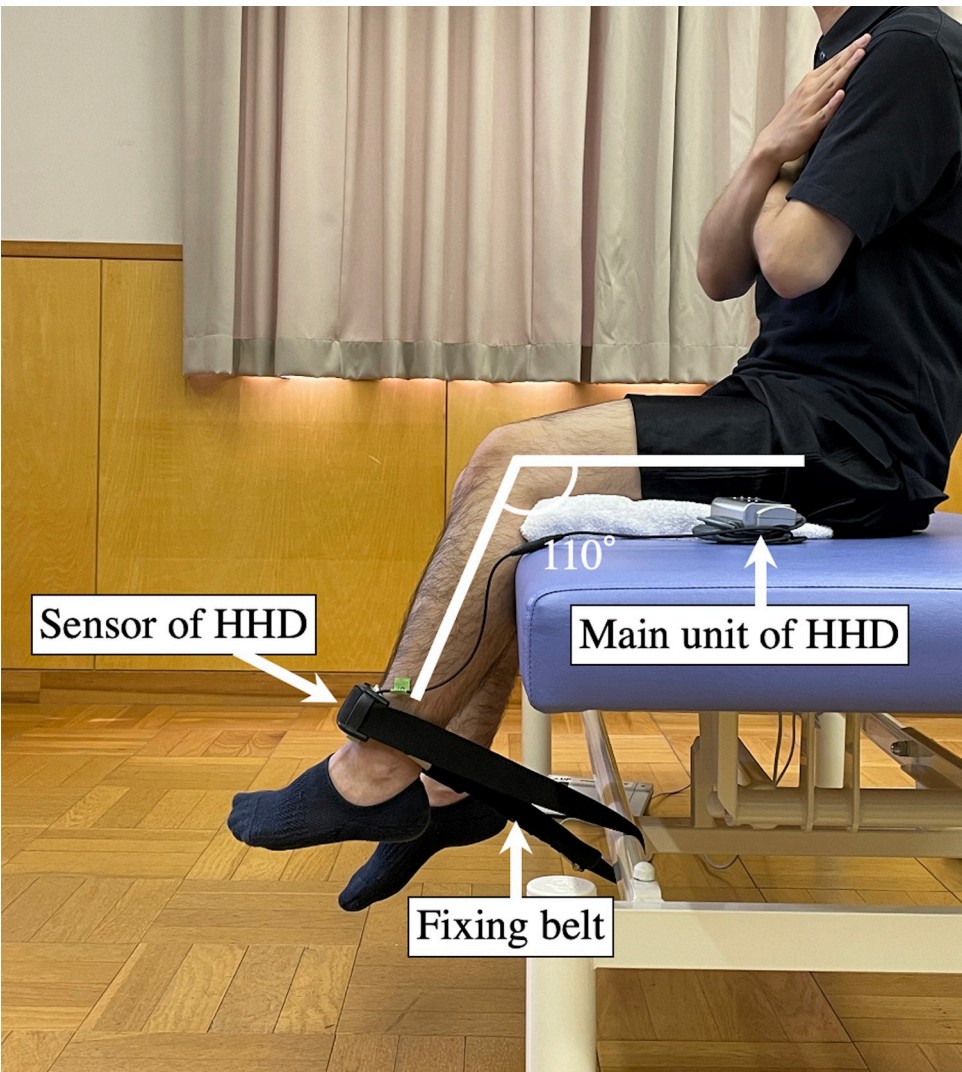

**Fig 2. Experimental setup for measuring the quadriceps muscle strength.**

Knee pain was measured with the visual analog pain scale (VAS). We asked the patient about the degree of pain during the measurement of gait speed. For pain intensity, the scale was anchored by "no pain" (score of 0 mm) and "worst possible pain" (score of 100 mm).

## Statistical analyses

Descriptive statistics were calculated for all variables and all continuous variables are expressed as means and standard deviations (SD). Only sex was shown in percentages. Pearson's correlation coefficients were used to assess the relationship between quadriceps strength, knee extension velocity, and gait function (gait speed and TUG) preoperatively and at 2 and 3 weeks following TKA. Pearson correlation coefficients were classified using the following definitions: 0–0.19, very weak; 0.2–0.39, weak; 0.40–0.59, moderate; 0.60–0.79, strong; and 0.80–1.0, very strong [30]. Multiple linear regression analysis with forced entry was performed with gait speed and TUG at each time as the dependent variables and age, sex, height, weight, pain, knee extension velocities of both sides, and quadriceps strength of both sides as independent

**Table 1. Preoperative characteristics of the patients.**

| | Male (n = 43) | | | | Female (n = 143) | | | | All (n = 186) | | | |
|---|---|---|---|---|---|---|---|---|---|---|---|---|
| | Value | | Range | | Value | | Range | | Value | | Range | |
| Age (years) | 75.8 ± 7.4 | | 62 – 92 | | 75.9 ± 6.4 | | 60 – 90 | | 75.9 ± 6.6 | | 60 – 92 | |
| Height (cm) | 161.1 ± 7.8 | | 148 – 176 | | 148.2 ± 5.7 | | 136 – 161 | | 151.2 ± 8.3 | | 136 – 176 | |
| Weight (kg) | 67.0 ± 10.7 | | 46.0 – 90.2 | | 58.0 ± 8.9 | | 36.7 – 83.5 | | 60.1 ± 10.0 | | 36.7 – 90.2 | |
| BMI (kg/m²) | 25.8 ± 3.8 | | 18.1 – 34.8 | | 26.5 ± 4.0 | | 17.3 – 47.8 | | 26.3 ± 3.9 | | 17.3 – 47.8 | |

Mean ± SD, BMI: Body Mass Index

variables. We calculated the variance inflation factor (VIF) to check for multicollinearity, and any independent variables for which VIF was greater than 5 were removed from the model [31]. All analyses were performed using SPSS Statistics version 25 (SPSS Japan, Inc., Tokyo), and P values <0.05 were considered significant.

## Results

A total of 186 patients participated in this study. The operation was performed by 14 surgeons at four hospitals, using a medial parapatellar approach in 182 cases and mini-midvastus approach in four cases. The type of implant utilized was posterior-stabilized in 121 cases, rotational platform in 30 cases, posterior cruciate-retaining in 17 cases, posterior cruciate-substituting in two cases, and unknown in 16 cases. All patients underwent a standard rehabilitation program per institutional protocol with a physical therapist 5 days a week for 3 weeks the day after TKA. Table 1 shows the baseline characteristics of the patients. The mean age of the patients was 75.9 ± 6.6 years, and 23.1% were male.

Table 2 shows the changes in gait and knee functions. The average gait speed declined at 2 weeks following TKA, but it had almost recovered to the preoperative levels at 3 weeks postoperatively. However, the patients demonstrated a decrease of over 30% in quadriceps strength on the operated side and a reduction in knee extension velocity of approximately 10% on the operated side even at 3 weeks after TKA.

Table 3 shows the correlations between knee extension velocity, quadriceps strength, and gait function. Both bilateral knee extension velocities and bilateral muscle strengths showed

**Table 2. Changes in gait and knee function over time (n = 186).**

| | PRE | | | 2 Weeks | | | 3 Weeks | | |
|---|---|---|---|---|---|---|---|---|---|
| Gait speed (m/s) | 1.14 | ± | 0.37 | 1.01 | ± | 0.28 | 1.12 | ± | 0.29 |
| Timed Up and Go (TUG) test (s) | 13.3 | ± | 5.0 | 13.8 | ± | 4.0 | 12.4 | ± | 3.3 |
| Knee extension velocity (°/s) | | | | | | | | | |
| (Operated side) | 425.5 | ± | 121.8 | 340.9 | ± | 98.3 | 373.7 | ± | 95.7 |
| (Non-operated side) | 472.8 | ± | 122.1 | 487.6 | ± | 128.0 | 499.1 | ± | 136.1 |
| Quadriceps strength (kg) | | | | | | | | | |
| (Operated side) | 14.7 | ± | 6.6 | 7.8 | ± | 3.3 | 9.4 | ± | 4.0 |
| (Non-operated side) | 18.1 | ± | 7.2 | 18.2 | ± | 6.6 | 18.2 | ± | 6.5 |
| Visual analog scale | 36.2 | ± | 25.6 | 27.2 | ± | 19.6 | 19.6 | ± | 17.1 |
| Range of motion (Operated side) (°) | | | | | | | | | |
| Extension | -9.2 | ± | 7.2 | -4.1 | ± | 4.2 | -3.2 | ± | 3.8 |
| Flexion | 122.7 | ± | 15.5 | 114.2 | ± | 12.2 | 119.6 | ± | 10.4 |

Mean ± SD

**Table 3. Correlations between gait function and knee function (n = 186).**

| | | Knee extension velocity | | Knee extension strength | |
|---|---|---|---|---|---|
| | | operated side | non-operated side | operated side | non-operated side |
| Pre-operation | Gait speed | 0.47 * | 0.32 * | 0.52 * | 0.52 * |
| | TUG | -0.54 * | -0.37 * | -0.46 * | -0.39 * |
| 2 weeks | Gait speed | 0.47 * | 0.28 * | 0.32 * | 0.40 * |
| | TUG | -0.49 * | -0.35 * | -0.38 * | -0.34 * |
| 3 weeks | Gait speed | 0.49 * | 0.31 * | 0.30 * | 0.41 * |
| | TUG | -0.48 * | -0.40 * | -0.33 * | -0.35 * |

*P < 0.001, TUG: Timed Up and Go test

weak-to-moderate correlations with gait speed and TUG at all time points. On the operated side, the correlation coefficients between the knee extension velocity and gait function (gait speed and TUG) were higher than those between the quadriceps strength and gait function at the postoperative time points.

Table 4 shows the results of multiple regression analysis. As all VIF values were lower than 5, we confirmed the absence of multicollinearity in both models. Knee extension velocity on the operated side, quadriceps strength on the non-operated side, and VAS were independently associated with gait speed at all time points. Knee extension velocity on the operated side was the most critical determinant of gait speed at all time points.

Concerning TUG, knee extension velocity on the operated side was the most significant determinant at all time points. Quadriceps strength on the non-operated side and VAS were also significant determinants at 2 weeks postoperatively.

## Discussion

The primary purpose of this study was to investigate the importance of knee extension velocity in determining gait function following TKA through comparison with quadriceps strength. The knee extension velocity on the operated side was found to be more strongly associated with gait function than quadriceps strength in the correlation analysis. It was the most important independent predictor of gait function from preoperative to 3 weeks postoperative in regression analysis. These results indicate that knee extension velocity is a more significant determinant of gait function than quadriceps strength on the operated limb during the early postoperative period following TKA.

To the best of our knowledge, this is the first investigation to measure the knee extension velocity and demonstrate the relationship between the velocity and gait function in patients with TKA. This study measured both velocity and strength on the same joint and direction (knee extension). We measured knee extension velocity without an external load to measure the velocity component as purely as possible, based on the force-velocity relationship [32]. Furthermore, this study was designed as a prospective and multi-institutional study. Therefore, we believe that the present results accurately reflect the importance of knee extension velocity for gait function in the early postoperative period following TKA.

There are two possible reasons why knee extension velocity is a significant determinant of gait function following TKA. Firstly, we focused on the relationship between the present study's knee extension velocity and the knee extension angular velocity during gait. Mentiplay et al. examined the angular velocities of lower limb joints during gait in a healthy population at various gait speeds (0.40–1.60 m/s). They showed the mean knee extension angular velocity was 357.9˚/s at 1.00–1.19 m/s in gait speed [33]. In the present study, gait speed was 1.01 m/s

**Table 4. Multiple linear regression analysis for maximum gait speed and TUG.**

| | | Dependent variable | | | | | |
|---|---|---|---|---|---|---|---|
| | | Gait Speed | | | TUG | | |
| | Independent variable | adjusted β | P-value | VIF | adjusted β | P-value | VIF |
| Pre-operation | Knee extension velocity (Operated side) | 0.318 | < 0.001 | 2.193 | -0.441 | < 0.001 | 2.193 |
| | Knee extension velocity (Non-operated side) | -0.034 | 0.684 | 2.031 | 0.028 | 0.746 | 2.031 |
| | Quadriceps strength (Operated side) | 0.108 | 0.343 | 3.821 | -0.213 | 0.069 | 3.821 |
| | Quadriceps strength (Non-operated side) | 0.291 | 0.012 | 3.920 | -0.083 | 0.481 | 3.920 |
| | VAS | -0.137 | 0.023 | 1.053 | 0.013 | 0.830 | 1.053 |
| | | adjusted R$^2$ = 0.383 | | | adjusted R$^2$ = 0.350 | | |
| 2 weeks | Knee extension velocity (Operated side) | 0.417 | < 0.001 | 1.925 | -0.306 | <0.001 | 1.925 |
| | Knee extension velocity (Non-operated side) | -0.117 | 0.145 | 1.798 | 0.012 | 0.873 | 1.798 |
| | Quadriceps strength (Operated side) | -0.068 | 0.395 | 1.806 | -0.108 | 0.169 | 1.806 |
| | Quadriceps strength (Non-operated side) | 0.301 | 0.002 | 2.583 | -0.214 | 0.023 | 2.583 |
| | VAS | -0.168 | 0.009 | 1.138 | 0.185 | 0.003 | 1.138 |
| | | adjusted R$^2$ = 0.350 | | | adjusted R$^2$ = 0.379 | | |
| 3 weeks | Knee extension velocity (Operated side) | 0.448 | < 0.001 | 1.816 | -0.340 | < 0.001 | 1.816 |
| | Knee extension velocity (Non-operated side) | -0.086 | 0.282 | 1.817 | -0.064 | 0.442 | 1.817 |
| | Quadriceps strength (Operated side) | -0.150 | 0.076 | 1.996 | -0.06 | 0.493 | 1.996 |
| | Quadriceps strength (Non-operated side) | 0.304 | 0.003 | 2.815 | -0.159 | 0.128 | 2.815 |
| | VAS | -0.146 | 0.028 | 1.236 | -0.023 | 0.743 | 1.236 |
| | | adjusted R$^2$ = 0.350 | | | adjusted R$^2$ = 0.295 | | |

Adjusted for age, sex, height, and weight, TUG: Timed Up and Go test, VAS: Visual analog scale, VIF: Variance inflation factor

at 2 weeks and 1.12 m/s at 3weeks after TKA and the knee extension velocities on the operated side in a seated position were 340.9˚/s at 2 weeks and 373.7˚/s at 3 weeks. These results indicate that the knee extension velocity in the sitting position of TKA patients whose gait speeds have recovered to 1.0–1.1 m/s is close to the knee extension angular velocity of healthy people during gait at a speed of 1.0–1.2 m/s. The proximity of the two values may partly explain why knee extension velocity measured in the sitting position was moderately correlated with gait speed.

Secondly, we focused on the common mechanism to determine knee motion during gait and knee extension movement during the velocity measurement. The kinematic characteristics of gait in patients with TKA have reduced knee flexion during the loading response, reduced knee extension during the late stance phase, and decreased knee flexion during the swing phase [34, 35]. In other words, the range of knee motion during gait is smaller than that in healthy elderly adults, and it has been called "stiff knee gait" [36–38]. Stiff knee gait leads to

shortening of the step length and ultimately slower gait speed [8]. One of the main etiologies of stiff knee gait is quadriceps/hamstrings co-activation [36, 37], and thus quadriceps/hamstrings co-activation can be assumed to have an effect on gait speed. Conversely, increased muscle co-activation also contributes to decreased maximum movement velocity in a single joint [39], so quadriceps/hamstrings co-activation may be a limiting factor for knee extension velocity during measurement in patients with TKA. From the above, quadriceps/hamstrings co-activation is considered a common determinant of both knee motion during gait and knee extension velocity during measurement. Furthermore, pain or/and fear of pain can also be common limitations of gait speed and knee extension velocity. We consider these commonalities an underlying mechanism of the significant relationship between gait function and knee extension velocity following TKA. Furthermore, it could be said that a decrease in knee extension velocity is directly related to decrease in gait speed.

In contrast, the knee extension velocity on the non-operated side was significantly but weakly associated with gait function and not selected as an independent variable in the regression analysis. This result indicated that the importance of knee extension velocities on the operated and non-operated side was different early after TKA. Although the loading condition was different from the present study, Pojednic et al. showed that the maximal knee extension velocity of a 40% one repetition-maximum leg press was a more critical determinant in mobility-limited than in healthy older adults [27]. Therefore, we speculate that the importance of movement velocity would increase with functional decline in the lower limb.

In addition to the knee extension velocity on the operated side, the quadriceps strength on the non-operated side is also a good determinant of gait function. This result of the importance of quadriceps strength is consistent with previous studies [40, 41]. Because quadriceps strength on the non-operated side, which reflects the progression of knee OA on the non-operated side [41], may have the ability to compensate for that on the operated side during gait [42], we suspect there was a significant relationship with gait function.

Based on the findings of the present study, early rehabilitation programs after TKA surgery may be recommended in order to focus on the knee extension velocity on the operated side and the quadriceps strength on the non-operated side. A previous study demonstrated that training without external load was the most effective factor in improving the maximum movement velocity [43]. Therefore, knee extension velocity training on the operated side without external load may be an effective training exercise. Furthermore, strength training involving the quadriceps muscle on the non-operated side may also improve gait function.

This study had two main limitations. First, the observation period for the knee and gait function following TKA was short (≧ 3 weeks). Therefore, we could not observe long-term changes. Second, kinematic and kinetic analyses during gait were not performed. To identify the reasons for the close association between the knee extension velocity of knee extension and gait function, further research, including detailed gait analysis, is needed in patients who underwent TKA.

In conclusion, knee extension velocity on the operated side was more closely associated with gait function than quadriceps strength in the early postoperative period following TKA in this prospective observational and multicenter study. The findings suggest that knee extension velocity has different characteristics from muscle strength, and knee extension velocity training may be considered a part of the rehabilitation program in the early postoperative period following TKA. However, we only had a relatively short postoperative observation period of 3 weeks; therefore, further research regarding interventions and long-term observations is needed.

## Supporting information

**S1 Dataset.**
(XLSX)

## Author Contributions

**Conceptualization:** Akira Iwata, Yuki Sano, Junji Inoue, Saki Yamamoto, Hiroshi Iwata.

**Data curation:** Yuki Sano, Hideyuki Wanaka, Shingo Kobayashi, Kensuke Okamoto, Jun Yamahara, Masaki Inaba, Yuya Konishi.

**Formal analysis:** Saki Yamamoto.

**Funding acquisition:** Akira Iwata.

**Investigation:** Hideyuki Wanaka, Junji Inoue, Saki Yamamoto.

**Methodology:** Akira Iwata, Yuki Sano, Hideyuki Wanaka, Shingo Kobayashi, Kensuke Okamoto, Jun Yamahara, Masaki Inaba, Yuya Konishi, Junji Inoue, Saki Yamamoto.

**Project administration:** Akira Iwata.

**Supervision:** Hiroshi Iwata.

**Writing – original draft:** Akira Iwata.

**Writing – review & editing:** Akira Iwata, Atsuki Kanayama, Hiroshi Iwata.

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
