## [Decision Letter · Decision Letter 0]

2 Mar 2022

PONE-D-21-17743

Maximum knee extension velocity without external load is a stronger determinant of gait function than quadriceps strength in the early postoperative period following total knee arthroplasty

PLOS ONE

Dear Dr. Iwata,

Thank you for submitting your manuscript to PLOS ONE. After careful consideration, we feel that it has merit but does not fully meet PLOS ONE’s publication criteria as it currently stands. Therefore, we invite you to submit a revised version of the manuscript that addresses the points raised during the review process.

The manuscript has been evaluated by two reviewers, and their comments are available below.

The reviewers have raised a number of major concerns. They request improvements to the reporting of methodological aspects of the study and questions regarding your statistical analysis. 

Could you please carefully revise the manuscript to address all comments raised?

We look forward to receiving your revised manuscript.

Kind regards,

Thomas Phillips, PhD

Associate Editor

PLOS ONE

Journal Requirements:

2. Thank you for stating the following financial disclosure: "This work was supported by JSPS KAKENHI Grant Number 20K11162. " ext-link-type="uri" xlink:type="simple">https://kaken.nii.ac.jp/grant/KAKENHI-PROJECT-20K11162/"

3. Please expand the acronym “JSPS” (as indicated in your financial disclosure) so that it states the name of your funders in full.

5. Please amend the manuscript submission data (via Edit Submission) to include author Atsuki Kanayama.

Reviewers' comments:

Reviewer's Responses to Questions

**Comments to the Author**

1. Is the manuscript technically sound, and do the data support the conclusions?

Reviewer #1: Partly

Reviewer #2: No

2. Has the statistical analysis been performed appropriately and rigorously? 

Reviewer #1: Yes

Reviewer #2: I Don't Know

3. Have the authors made all data underlying the findings in their manuscript fully available?

Reviewer #1: Yes

Reviewer #2: Yes

4. Is the manuscript presented in an intelligible fashion and written in standard English?

Reviewer #1: Yes

Reviewer #2: No

5. Review Comments to the Author

Reviewer #1: This is an interesting study about TKR and Knee extension velocity. The study is well written, with for me, only some remarks.

*Introduction

Well written-

Please add references for these two sentences: “However, in clinical situations, there are some patients whose gait function recovers to normal levels despite a considerable reduction in quadriceps strength. Conversely, some patients may have recovered to sufficient muscle strength but still have a decline in gait function. »

Please Delete this sentence : « (muscle power = muscle strength × movement velocity) »

Please add reference for this sentence : « Movement velocity is the speed at which the upper/lower limbs or trunk are moved as quickly as possible under various loading conditions. »

*Methods

Measurements:

I think that it could be important to have a description of the type of implant used and surgery approach used also and the number of surgeons because there is a real impact on TKA, recovery, quadriceps strength …. We know nothing about this.

In addition, it is also important to explain the rehabilitation session received by the patient after TKA. As surgery technique and type of implant, there is a large impact of this rehabilitation on recovery especially after such a short time after TKA.

Add BMI in your list. Maybe it is sufficient to have only BMI and height and weight----

Please define TUG and add references to justify that it is a good test to evaluate gait function.

Please add for ROM, that it is a PASSIVE ROM….. indeed, we can think of the active measurement made during walking.

Please add the units for all your parameters used in your manuscript.

Please add reference to justify the nearest of 5° using the goniometer.

Statistical section:

Please add that you also used percentages of the description of your cohort. Define also SD.

Concerning the correlations, please add a reference to explain the strength of the link between the data.

Did you test uni-regression before to do your model? How you justify the choice of your co-variate in your model? Did you test the correlation between each co-variates before your final model?

*Results

Regarding the range of your BMI, you have a large range and very little SD, did you cheeked your data to explain this important difference?

Table 3, please precise in your results that the correlation are indeed significant but always moderate or low … it is important to indicate this information, to always think in terms of clinical approach.

Table 4 :

*Discussion

This section is well written.

Please modify this sentence because we think that the knee extension velocities have been recorded directly during the gait with reflective markers for examples : “In the present study, the maximum knee extension velocities on the operated side were 340.9 °/s at 1.01 m/s in gait speed (2 weeks) and 373.7 °/s at 1.12 m/s (3 weeks).“

In the sentence below, I think that it is important to remain cautious in the statements made. This may indeed be an important factor, but the angular velocity during operation depends on many other factors.

« This closeness may indicate that the knee extension velocity is a limiting factor of gait speed. This factor may be one reason for the strong relationship between knee extension velocity and gait speed.“

Concerning the stiff knee, there are several mechanics that can cause this inefficient gait. The first is mainly caused by inappropriate swing-phase activity of RF (necessary to have EMG) referred as spasticity (stroke, CP patients…). The second mechanism may be the abnormal foot kinematic at toe-off with a lack of push-off power at the ankle joint with a consequent lack of passive knee flexion. The third mechanism may be the insufficient hip pull-off that should produce hip flexion during swing…. Finally, there is an important problem of compensation gait for this population of patient with OA during several years…. And also an important impact of the pain or fear of pain with therefore a slow gait and compensation…

Maybe it is important to precise this in your manuscript.

Reviewer #2: I have strong reservation on the methods. Not much details of the measurement and data processing techniques are provided in this paper.

1. The authors rightly pointed out that muscle power (a produce of strength and velocity) is important in clinical rehabilitation. I wonder why the authors did not examine power but single out the velocity component. Can they compare power, strength and velocity? Rather than advocating one better than another, will the variables collectively inform the clinicians better?

2. Handhold dynamometer is not the best for knee extension testing as it is hard to stabilize the joint. Did the authors check the joint angles during the tests using objective methods? It is likely that the leg will move away from the initial start position of 70 degrees of knee flexion.

3. What is the reliability of the equipment and testers in this study?

4. I was under the impression that the knee extension strength was measured as isometric contraction (knee flexion angle fixed at 70 degrees as much as possible). How did the authors measure velocity for isometric extension using a handhold dynamometer? This part is not clear.

5. Can a picture be provided to illustrate the test protocol? There was a figure at the end but rather strange. The angle there definitely

6. Please prove more details on how raw data are processed and how each variables (e.g. velocity) are calculated or extracted.

7. I am unsure about the data in Table 2. Knee extension velocity of over 400 degrees per second – during what movement? How?

8. I did not read further than the results because I have some doubt about the methodology and data integrity.

6. PLOS authors have the option to publish the peer review history of their article (what does this mean?). If published, this will include your full peer review and any attached files.

Reviewer #1: **Yes: **Alice BONNEFOY-MAZURE, PhD

Reviewer #2: No

---

## [Author Response · Author response to Decision Letter 0]

20 Jun 2022

Journal Requirements:

1. We have ensured to satisfy PLOS ONE’s style requirements and have revised our manuscript.

2.3. This work was supported by Japan Society for the Promotion of Science KAKENHI Grant Numbers 20K11162. The funders had no role in study design, data collection, and analysis, decision to publish or preparation of the manuscript.

4. We have uploaded our study’s data set to Supporting Information.

5. We amended the submission data and included author Atsuki Kanayama.

Reviewer #1: This is an interesting study about TKR and Knee extension velocity. The study is well written, with for me, only some remarks.

−We thank the reviewer for these positive comments. 

*Introduction

Please add references for these two sentences: “However, in clinical situations, there are some patients whose gait function recovers to normal levels despite a considerable reduction in quadriceps strength. Conversely, some patients may have recovered to sufficient muscle strength but still have a decline in gait function. »

−We appreciate the reviewer’s comment on this point. We have described these sentences based on our clinical experience. However, we cannot add references to these. Thus, we have deleted this part and changed it.

Please Delete this sentence : « (muscle power = muscle strength × movement velocity) »

−As suggested, we deleted this sentence.

Please add reference for this sentence : « Movement velocity is the speed at which the upper/lower limbs or trunk are moved as quickly as possible under various loading conditions. »

−Thank you for your suggestion. We have added the references.

*Methods

Measurements:

I think that it could be important to have a description of the type of implant used and surgery approach used also and the number of surgeons because there is a real impact on TKA, recovery, quadriceps strength …. We know nothing about this. 

In addition, it is also important to explain the rehabilitation session received by the patient after TKA. As surgery technique and type of implant, there is a large impact of this rehabilitation on recovery especially after such a short time after TKA.

−Thank you for pointing this out. We have added a description about the number of surgeons, surgical approach, type of implants, and rehabilitation program at the beginning of the Results section.

Add BMI in your list. Maybe it is sufficient to have only BMI and height and weight--

−This has been revised as suggested by the reviewer.

Please define TUG and add references to justify that it is a good test to evaluate gait function.

−As suggested by the reviewer, we added a sentence.

Please add for ROM, that it is a PASSIVE ROM….. indeed, we can think of the active measurement made during walking.

−As the reviewer noted, our original expression was confusing. We clarified that it was a passive ROM. 

Please add the units for all your parameters used in your manuscript.

−As the reviewer suggested, we have added the units for all parameters in the Measurements section.

Please add reference to justify the nearest of 5° using the goniometer.

−Thank you for your suggestion. We have added the reference.

Statistical section:

Please add that you also used percentages of the description of your cohort. Define also SD.

−Thank you for emphasizing this. I have added a description about using percentage and the definition of SD.

Concerning the correlations, please add a reference to explain the strength of the link between the data.

−In accordance with the reviewer's comment, we have added an explanation about the strength of correlation coefficient and a reference.

Did you test uni-regression before to do your model? How you justify the choice of your co-variate in your model? Did you test the correlation between each co-variates before your final model?

−We thank the reviewer for this insightful comment. This study aimed to identify the importance of the knee extension velocity in determining the gait function after TKA. Therefore, we need to enter bilateral quadriceps strength and pain as covariates in the same model, along with knee extension velocity. As the reviewer pointed out, it is necessary to check for multicollinearity; hence, we added to the Statistical analyses section and Table 4.

*Results

Regarding the range of your BMI, you have a large range and very little SD, did you cheeked your data to explain this important difference?

−Thank you for pointing this out. We checked our data again and there were no errors. We provide the histogram of BMI.

Table 3, please precise in your results that the correlation are indeed significant but always moderate or low … it is important to indicate this information, to always think in terms of clinical approach.

−We appreciate the reviewer’s comment. We have added a description about the strength of the correlation coefficient in the Results section.

*Discussion

This section is well written.

Please modify this sentence because we think that the knee extension velocities have been recorded directly during the gait with reflective markers for examples : “In the present study, the maximum knee extension velocities on the operated side were 340.9 °/s at 1.01 m/s in gait speed (2 weeks) and 373.7 °/s at 1.12 m/s (3 weeks).“

−As the reviewer noted, our original expression was confusing. To avoid confusion, we have shown the results for the knee extension velocity and gait speed separately, and knee extension velocity during gait and sitting was clearly described.

In the sentence below, I think that it is important to remain cautious in the statements made. This may indeed be an important factor, but the angular velocity during operation depends on many other factors.

« This closeness may indicate that the knee extension velocity is a limiting factor of gait speed. This factor may be one reason for the strong relationship between knee extension velocity and gait speed.“

−We agree with the reviewer's comment that we should express our thoughts more cautiously. We have changed this part as follows.

Concerning the stiff knee, there are several mechanics that can cause this inefficient gait. The first is mainly caused by inappropriate swing-phase activity of RF (necessary to have EMG) referred as spasticity (stroke, CP patients…). The second mechanism may be the abnormal foot kinematic at toe-off with a lack of push-off power at the ankle joint with a consequent lack of passive knee flexion. The third mechanism may be the insufficient hip pull-off that should produce hip flexion during swing…. Finally, there is an important problem of compensation gait for this population of patient with OA during several years…. And also an important impact of the pain or fear of pain with therefore a slow gait and compensation…

Maybe it is important to precise this in your manuscript.

−Thank you very much for this insightful comment. We recognize and agree with this comment that the stiff knee is caused by inappropriate activity in RF, lack of push-off, and insufficient hip pull-off. Furthermore, we have been made aware by the reviewer's comment that the pain or fear of pain can be a common limiting factor for gait speed and knee extension velocity. Therefore, we have added a sentence about pain or/and fear of pain.

We wish to thank the Reviewer again for his/her valuable comments.

 

Responses to Reviewer #2:

−We thank you for the time and effort spent in reviewing our manuscript and suggesting some important points to consider.

−First of all, the representation of the knee angle was confusing because we used mixed expressions for the knee flexion angle and knee joint angle. In the revised submission, we have unified the use of knee joint angles. Additionally, we stated the endpoint for measuring knee extension velocity as 170° of knee extension; however, this was an error of 160° in a knee joint angle. We revised our manuscript as follows.

I have strong reservation on the methods. Not much details of the measurement and data processing techniques are provided in this paper.

1. The authors rightly pointed out that muscle power (a produce of strength and velocity) is important in clinical rehabilitation. I wonder why the authors did not examine power but single out the velocity component. Can they compare power, strength and velocity? Rather than advocating one better than another, will the variables collectively inform the clinicians better?

−As noted by the reviewer, we designed this study to confirm the importance of the velocity component to gait function in the early postoperative period following TKA. The reason for setting this objective is that the standard rehabilitation program focuses on the strength component but not the velocity component. Since the importance of power and strength components to gait function has been compared in previous studies (Bean JF, 2003), we thought it would be possible to compare the velocity component to muscle strength. 

As the reviewer pointed out, it is certainly not easy to separate the muscular and velocity component completely. Therefore, to separate each component as possible, strength was measured with the velocity set to 0 (isometric contraction), and velocity was measured with the external load set to 0. Consequently, the importance of the velocity component to gait function was shown in this study. We believe that considering the velocity component in rehabilitation programs may be useful in restoring gait function in the early postoperative period following TKA.

2. Handhold dynamometer is not the best for knee extension testing as it is hard to stabilize the joint. Did the authors check the joint angles during the tests using objective methods? It is likely that the leg will move away from the initial start position of 70 degrees of knee flexion.

−We used a hand-held dynamometer (HDD) equipped with a fixing belt (μ-Tas F-1, Anima Corp., Tokyo, Japan). We show two figures from previous studies using this HDD (Katoh M, 2009 and Kiyohara M). Using a belt improved the reliability of isometric strength measurement using HDD, which has proven to be sufficiently reliable (Katoh M, 2009). In this method, the joint angle can be set by adjusting the length of the belt. In the present study, we have checked the knee joint angle and adjusted the length of the belt to set 110° in one practical trial.

As pointed out by the reviewer, there was a lack of explanation regarding the measurement of isometric muscle force. In addition, the text “The patient was seated with their knees positioned at 70° flexion.” was also misleading. Therefore, we have added the explanation.

3. What is the reliability of the equipment and testers in this study?

−We thank the reviewer for this comment. We have preliminarily verified the inter- and the intra-rater reliability of knee extension velocity and quadriceps muscle strength in young people (n=11). The intraclass correlation coefficient (ICC) was used to evaluate the reliability. As a result, the reliability showed a high-class correlation coefficient in almost all values. Therefore, we used both tests in this study.

4. I was under the impression that the knee extension strength was measured as isometric contraction (knee flexion angle fixed at 70 degrees as much as possible). How did the authors measure velocity for isometric extension using a handhold dynamometer? This part is not clear.

−We appreciate the reviewer's comment on this point. We separately measured "knee extension velocity" and "quadriceps isometric strength" in this study. We used a gyro-sensor to measure the velocity when the patient extended their knee as fast as possible from a knee joint angle of 90° to 160° and a hand-held dynamometer to measure the strength at knee joint angle of 110°. To make this point clearer, we have revised Figure 1 (velocity) and added Figure 2 (strength).

5. Can a picture be provided to illustrate the test protocol? There was a figure at the end but rather strange. The angle there definitely

−In accordance with the reviewer's comment, we have changed Figure 1 and added Figure 2.

6. Please prove more details on how raw data are processed and how each variables (e.g. velocity) are calculated or extracted.

−We measured knee extension velocity using a wireless gyro-sensor. The sampling rate of the gyro-sensor was 200 Hz and the measurement data were transferred to a personal computer. As suggested, we have added these detailed information to the knee function section.

7. I am unsure about the data in Table 2. Knee extension velocity of over 400 degrees per second – during what movement? How?

−We measured the knee extension angular velocity using a gyro-sensor when patients with TKA extended their knee as fast as possible from a knee joint angle of 90° to 160° in the sitting position. The knee extension velocity was measured in 113 healthy elderly subjects with an average age of 72.5 years (599.2 ± 108.4°/s [unpublished]). Considering the age and the velocity values, we believe that the data from the TKA patients in the present study are accurate.

8. I did not read further than the results because I have some doubt about the methodology and data integrity.

−Again, we deeply appreciate the reviewer for her/his time on our study and we apologize for the confusion. As mentioned above, we exerted all efforts to clarify the methods in detail. Therefore, now, we deem that it is much clearer for the readers.

---

## [Decision Letter · Decision Letter 1]

16 Aug 2022

PONE-D-21-17743R1Maximum knee extension velocity without external load is a stronger determinant of gait function than quadriceps strength in the early postoperative period following total knee arthroplastyPLOS ONE

Dear Dr. Iwata,

Thank you for submitting your manuscript to PLOS ONE. After careful consideration, we feel that it has merit but does not fully meet PLOS ONE’s publication criteria as it currently stands. Therefore, we invite you to submit a revised version of the manuscript that addresses the points raised during the review process. Can you please address the outstanding concerns raised by the reviewer?

We look forward to receiving your revised manuscript.

Kind regards,

Avanti Dey, PhD

Staff Editor

PLOS ONE

Journal Requirements:

Reviewers' comments:

Reviewer's Responses to Questions

**Comments to the Author**

1. If the authors have adequately addressed your comments raised in a previous round of review and you feel that this manuscript is now acceptable for publication, you may indicate that here to bypass the “Comments to the Author” section, enter your conflict of interest statement in the “Confidential to Editor” section, and submit your "Accept" recommendation.

Reviewer #2: (No Response)

2. Is the manuscript technically sound, and do the data support the conclusions?

Reviewer #2: Yes

3. Has the statistical analysis been performed appropriately and rigorously? 

Reviewer #2: N/A

4. Have the authors made all data underlying the findings in their manuscript fully available?

Reviewer #2: Yes

5. Is the manuscript presented in an intelligible fashion and written in standard English?

Reviewer #2: Yes

6. Review Comments to the Author

Reviewer #2: The authors put in a lot of effort to clarify my previous concerns. They also added more figures and text to better explain their procedures in the manuscript.

1. Did the authors use tolerance value in addition to VIF? What do they mean by “Multicollinearity may be present”? The word “maybe” is rather vague – do the authors use a cut-off value of VIF 5 to make some decision? Please be very clear on the interpretation and steps to take when VIF is greater than 5.

2. Table 1 – This table can be expanded to include data of males only, females only and all participants as a group. It will be good if other demographic characteristics (e.g. how many use a walking stick, medical history, time from operation) can be included if such data are available.

3. Line 248 – Please rephrase “This closeness”.

4. In the Discussion, the authors wrote one paragraph on “Secondly, we focused on the common mechanism to determine the knee extension velocity and the knee motion during gait….” However, after reading the entire paragraph which is primarily a summary of other studies, it is puzzling what the present study contributed to the mechanism. It would be better if the authors can relate the findings from their own study with the literature in a more explicit manner.

5. The authors demonstrated experimentally that knee extension velocity on the operated side is a better predictor of gait function than quadriceps strength. This is nice from a research perspective, but is there any practical implication? In clinical practice, it is a lot easier to measure gait speed and TUG than knee extension velocity or even quadriceps strength. Thus, it is unlikely that clinician will measure knee extension velocity. Are there any direct clinical implications from this study?

6. In response to the reviewers’ previous comments, the authors presented a figure on the velocity raw data from the gyro-sensor. It will be nice to incorporate this figure together with Figure 1 to comprehensively show to test procedures (start/end position, sensor, data).

7. PLOS authors have the option to publish the peer review history of their article (what does this mean?). If published, this will include your full peer review and any attached files.

Reviewer #2: No

---

## [Author Response · Author response to Decision Letter 1]

29 Sep 2022

Reviewer #2: The authors put in a lot of effort to clarify my previous concerns. They also added more figures and text to better explain their procedures in the manuscript.

Response: Thank you for acknowledging our effort. We would like to express our appreciation to the reviewer for providing these constructive suggestions that have helped us to greatly improve our manuscript. We have revised the manuscript in accordance with the reviewer’s suggestions. 

1. Did the authors use tolerance value in addition to VIF? What do they mean by “Multicollinearity may be present”? The word “maybe” is rather vague – do the authors use a cut-off value of VIF 5 to make some decision? Please be very clear on the interpretation and steps to take when VIF is greater than 5.

Response: We thank the reviewer for the insightful comment. We used the VIF to check for multicollinearity among the independent variables, with a VIF5 indicative of multicollinearity (Cohen J, 2013). If multicollinearity was observed, we assumed that the variables should be removed from the model. However, as none of the variables in any model used in this study showed a VIF of more than five, no variable was removed. We agree with the reviewer that our expression regarding this issue may be unclear, so we have changed it. Furthermore, as we did not mention VIF in the Results section, we have added a brief description.

2. Table 1 – This table can be expanded to include data of males only, females only and all participants as a group. It will be good if other demographic characteristics (e.g. how many use a walking stick, medical history, time from operation) can be included if such data are available.

Response: As suggested, we have added sex-specific data for males and females only to Table 1. Unfortunately, data regarding a walking stick, medical history, and time since surgery, were not available. We aim to collect this data in future studies. 

3. Line 248 – Please rephrase “This closeness”.

Response: As suggested by the reviewer, we have changed from “This closeness” to “The proximity of the two values”.

4. In the Discussion, the authors wrote one paragraph on “Secondly, we focused on the common mechanism to determine the knee extension velocity and the knee motion during gait….” However, after reading the entire paragraph which is primarily a summary of other studies, it is puzzling what the present study contributed to the mechanism. It would be better if the authors can relate the findings from their own study with the literature in a more explicit manner.

Response: Thank you for your insightful comment. As suggested, to relate our findings with those of previous studies, we have edited and reconstructed the Discussion section and added more detailed explanations.

5. The authors demonstrated experimentally that knee extension velocity on the operated side is a better predictor of gait function than quadriceps strength. This is nice from a research perspective, but is there any practical implication? In clinical practice, it is a lot easier to measure gait speed and TUG than knee extension velocity or even quadriceps strength. Thus, it is unlikely that clinician will measure knee extension velocity. Are there any direct clinical implications from this study?

Response: We understand the reviewer’s concern that measuring knee extension velocity using a gyro-sensor is more complicated than measuring gait speed, TUG, or even quadriceps strength. However, a recent study showed that angular velocity could be easily measured with a smartphone application, although there is limited validation to support its accuracy (Hughes GTG, 2021). Therefore, we believe that the measurement of knee extension velocity will become more accessible in the future, allowing it to be performed in clinical situations. 

Quadriceps strength is considered a primary determinant of gait function, particularly early after TKA. However, as noted in the Introduction, there is often a large gap between quadriceps weakness and the recovery of gait function. Our results revealed that knee extension velocity on the operative side is a more critical determinant of gait function than quadriceps strength, indicating that rehabilitation focusing on knee velocity may be more effective. Therefore, we plan a future study focusing on knee extension velocity. If significant improvements in gait function are achieved, we believe that there will be a paradigm shift in clinical rehabilitation after TKA.

6. In response to the reviewers’ previous comments, the authors presented a figure on the velocity raw data from the gyro-sensor. It will be nice to incorporate this figure together with Figure 1 to comprehensively show to test procedures (start/end position, sensor, data).

Response: As the reviewer suggested, we have summarized the setup for the measurement, sensor, and representative data in Figure 1.

---

## [Decision Letter · Decision Letter 2]

4 Oct 2022

Maximum knee extension velocity without external load is a stronger determinant of gait function than quadriceps strength in the early postoperative period following total knee arthroplasty

PONE-D-21-17743R2

Dear Dr. Iwata,

We’re pleased to inform you that your manuscript has been judged scientifically suitable for publication and will be formally accepted for publication once it meets all outstanding technical requirements.

Kind regards,

Emiliano Cè

Academic Editor

PLOS ONE

Additional Editor Comments (optional):

Reviewers' comments:

Reviewer's Responses to Questions

**Comments to the Author**

1. If the authors have adequately addressed your comments raised in a previous round of review and you feel that this manuscript is now acceptable for publication, you may indicate that here to bypass the “Comments to the Author” section, enter your conflict of interest statement in the “Confidential to Editor” section, and submit your "Accept" recommendation.

Reviewer #2: All comments have been addressed

2. Is the manuscript technically sound, and do the data support the conclusions?

Reviewer #2: Yes

3. Has the statistical analysis been performed appropriately and rigorously? 

Reviewer #2: Yes

4. Have the authors made all data underlying the findings in their manuscript fully available?

Reviewer #2: Yes

5. Is the manuscript presented in an intelligible fashion and written in standard English?

Reviewer #2: Yes

6. Review Comments to the Author

Reviewer #2: The authors have addressed all my comments adequately. I think this paper is a nice contribution to the literature.

7. PLOS authors have the option to publish the peer review history of their article (what does this mean?). If published, this will include your full peer review and any attached files.

Reviewer #2: No

---

## [Editor Report · Acceptance letter]

14 Nov 2022

PONE-D-21-17743R2 

Maximum knee extension velocity without external load is a stronger determinant of gait function than quadriceps strength in the early postoperative period following total knee arthroplasty 

Dear Dr. Iwata:

I'm pleased to inform you that your manuscript has been deemed suitable for publication in PLOS ONE. Congratulations! Your manuscript is now with our production department. 

Kind regards, 

on behalf of

Professor Emiliano Cè 

Academic Editor

PLOS ONE